# Lessons Learned from Active Clinical and Laboratory Surveillance during the Sheep Pox Virus Outbreak in Spain, 2022–2023

**DOI:** 10.3390/v16071034

**Published:** 2024-06-27

**Authors:** Rubén Villalba, Andy Haegeman, María José Ruano, María Belén Gómez, Cristina Cano-Gómez, Ana López-Herranz, Jesús Tejero-Cavero, Jaime Capilla, María Victoria Bascuñan, Nick De Regge, Montserrat Agüero

**Affiliations:** 1Laboratorio Central de Veterinaria (LCV), Ministry of Agriculture, Fisheries and Food, 28110 Algete, Spain; mruanor@mapa.es (M.J.R.); mgmartin@mapa.es (M.B.G.); ccano@mapa.es (C.C.-G.); alherranz@mapa.es (A.L.-H.); 2Sciensano, Infectious Diseases in Animals, Exotic and Vector-Borne Diseases, 1180 Brussels, Belgium; andy.haegeman@sciensano.be (A.H.); nick.deregge@sciensano.be (N.D.R.); 3Junta de Comunidades de Castilla-La Mancha, 45071 Toledo, Spain; jtejero@jccm.es (J.T.-C.); jcapilla@jccm.es (J.C.); mbascunanc@jccm.es (M.V.B.)

**Keywords:** sheep pox, Capripoxvirus, surveillance, real-time PCR, oral swab

## Abstract

In September 2022, more than 50 years after its eradication from Spain, Sheep pox virus was confirmed by laboratory analysis in sheep showing characteristic lesions. This was the start of an outbreak that lasted 9 months and infected 30 farms dispersed over two different areas, Andalusia and Castilla-La Mancha. Early after the initial confirmation, an active surveillance based on clinical inspection with laboratory confirmation of sheep with clinical signs was started in restricted areas. This allowed the confirmation of Sheep pox in 22 out of 28 suspected farms, where limited numbers of sheep with mainly erythema and papules were found, indicative of early detection. Nevertheless, to improve active surveillance and stop the outbreak, clinical inspection was reinforced by laboratory analysis in all inspected farms, even when no clinically diseased sheep were detected. Although more than 35,000 oral swabs from 335 farms were analysed by real-time PCR in pools of five, only two out of six reported outbreaks in this period were detected by laboratory analysis before clinical signs were observed. Furthermore, additional insights were gained from the extensive laboratory surveillance performed on samples collected under field conditions. No evidence of Sheep pox virus infection was found in goats. Oral swabs proved to be the sample of choice for early detection in the absence of scabs and could be tested in pools of five without extensive loss in sensitivity; serology by ELISA was not useful in outbreak detection. Finally, a non-infectious genome of the virus could be detected months after cleaning and disinfection; thus, real-time PCR results should be interpreted with caution in sentinel animals during repopulation. In conclusion, the outbreak of Sheep pox virus in Spain showed that active clinical inspection with laboratory confirmation of clinically diseased sheep via oral swab testing proved a sensitive method for detection of infected farms, providing insights in laboratory surveillance that will be helpful for other countries confronted with Sheep pox outbreaks.

## 1. Introduction

Sheep pox (SPP) is a World Organization of Animal health (WOAH) notifiable disease [1] caused by the Sheep pox virus (SPPV) belonging to the genus Capripoxvirus, together with Goat pox virus (GTPV) and Lumpy skin disease virus (LSDV) [2]. Influenced by a number of factors, such as virus strain as well as sheep breed and age, SPPV can cause severe clinical disease in sheep, although some strains can infect both sheep and goats [3]. A SPPV infection is characterised by fever, generalised papules or nodules, and sometimes vesicles or internal lesions, and death [4,5,6]. The main mode of SPPV transmission is by direct contact between animals. Since the virus is persistent in the environment, it can also be transmitted via indirect contact with fomites such as workers, vehicles, tools, clothes, trade of hides, etc. [5].

SPPV is endemic in many countries in Africa, the Middle East, and Asia. The virus was endemic in Europe in ancient times but was eradicated from France, Spain, and Portugal in 1967, 1968, and 1969, respectively [7]. SPP is currently considered exotic to the European Union (EU), although several outbreaks occurred in past decades, mostly in countries neighbouring the SPPV endemic region. For example, repeated outbreaks have been reported in Greece and Bulgaria since the 1990s [8]. According to EU legislation [9,10], as it is a category A disease, its detection requires measures for immediate eradication.

In Spain, SPPV was considered eradicated since 1968. This suddenly changed on 15 September 2022, when sheep with unusual skin lesions were observed on a sheep holding in the municipality of Benamaurel, in the province of Granada, Andalusia (AND). The clinical suspicion occurred in a breeding sheep holding for meat production with a census of 314 sheep and 11 goats. Skin lesions compatible with SPP or GTP were detected in 50 clinically affected sheep, including 30 recently deceased animals. The official veterinary services (OVS) were notified, and samples were collected and sent to the national reference laboratory (NRL) in Spain, LCV, which confirmed the presence of SPPV within 24 h. In the frame of the epidemiological investigation following the outbreak, sheep with SPPV-compatible lesions and symptoms were also detected in a farm in the municipality of Villaescusa de Haro, in Cuenca province, Castilla–La Mancha (C-LM), on 23 September 2022. This farm was located almost 300 km from the first outbreak in AND and had a census of 1070 sheep and 14 goats. The presence of SPPV was again confirmed by the NRL. This represented the start of a SPPV outbreak in Spain that lasted for 9 months, during which 30 infected farms were identified. 

After the notification of SPPV in two regions in Spain, passive surveillance was enhanced throughout the country to ensure the early detection and reporting of any case of skin lesions compatible with SPPV infection. In addition, an active surveillance system based on clinical inspection in sheep and goat farms located in the restricted zone around the outbreak or epidemiologically linked to affected holdings was implemented. In the case of clinical suspicion, different samples (serum, EDTA blood, oral swab, and/or scabs) allowing for virological and serological diagnosis were collected and sent to the NRL.

Since new infected herds continued to be detected despite this active surveillance and associated control measures, a reinforced active surveillance was installed between 14 November and 5 December 2022 in AND, and between 1 December 2022 and 27 April 2023 in C-LM. During these periods, samples (oral swabs) were collected during farm visits in the restricted zone even in the absence of clinical signs. Based on the obtained results, which are described in the current manuscript, it was decided to return to the normal active surveillance system after 27 April 2023. This was maintained until the final eradication (Figure 1).

This manuscript will report clinical and laboratory findings during the entire SPPV outbreak in Spain in 2022–2023 and discuss lessons learned for implementation of laboratory surveillance in the case of future outbreaks. Information is furthermore provided on the clinical and laboratory surveillance upon repopulation of cleaned and disinfected farms.

## 2. Materials and Methods

### 2.1. Clinical Inspection

The clinical inspection of animals carried out during the active surveillance followed a system whereby all animals on the farm were inspected. Each batch, made up of about 20 animals, was moved to check if any had difficulties in movement (due to fever, lameness, respiratory difficulty, etc.), with detection of possible skin lesions (erythema, papules, or scabs). In the case the lambs were lethargic, they were carefully observed for skin lesions, especially in the armpit area, groin, and base of the tail.

### 2.2. Sampling

Table 1 and Table 2 show the type of surveillance that was carried out, the total number of farms officially sampled by OVS, and the total number of samples collected in AND and C-LM from the beginning of the outbreak until the 4 July 2023. Depending on the farm, serum, EDTA blood, material from skin lesions (scabs), and/or oral swabs were collected. Two-hundred and twelve sheep from outbreak farms 2022/12, 15, 18, and 19 that showed no scabs were subjected to paired EDTA blood and oral swab sampling to determine whether one of both matrices was more suitable for early SPPV detection after infection. Seventeen sheep were sampled two times; therefore, a total of 229 paired samples were available. These samples are recorded in Table 1. Separate from the samples mentioned in Table 1 and Table 2, clinical inspection and sample collection was also done on fourteen farms that were repopulated months after SPPV-positive animals had been identified. A total of 561 serum samples, 942 oral swabs, and 10 environmental swabs were collected on those farms.

### 2.3. Preparation and Processing of Samples

#### 2.3.1. Oral Swabs

Dry oral swabs were collected from each animal. Upon arrival in the laboratory, the swabs were completely submerged in 1 mL of PBS and maintained for 1 h at rest at room temperature to guarantee complete elution of the material. Subsequently, they were stored at 2–8 °C before starting the viral DNA extraction procedure.

Besides the testing of individual oral swabs, the possibility of pooling swabs was assessed, and the impact of pooling of swabs on sensitivity was evaluated. Two positive farms (2022/15 and 18) were therefore resampled some days after the initial sampling, and oral swabs from 60 sheep from each farm were collected in duplicate (replicates A and B).

Based on the promising results as will be described below, this procedure of testing pooled swabs was applied along the reinforced active surveillance period.

#### 2.3.2. Peripheral Blood

Peripheral blood samples were collected using vacuum tubes, with and/or without EDTA. Whole blood samples were allowed to clot at room temperature to obtain serum that was stored at 2–8 °C until testing by ELISA. Also, EDTA blood samples were stored at 2–8 °C until laboratory analysis by real-time PCR (rPCR).

#### 2.3.3. Scabs from Skin Lesion

Scab samples from the first outbreak, 2022/1, were collected in tubes containing 1 mL of phosphate buffered saline (PBS). Scabs were removed from the PBS and treated as described in the paragraph below. Both the scabs and the PBS in which the swabs had been stored (2–8 °C) were tested in rPCR.

During the other outbreaks, scabs from skin lesions were collected and stored in dry tubes at 2–8 °C before testing. Scabs were cut into pieces of around 4 mm^2^ and ground with glass beads in 1 mL PBS using an automatic homogenizer. After centrifugation at 1000× *g* for 15 min at 4 °C, supernatants were transferred to a new tube and stored at +4 °C until testing.

### 2.4. Nucleic Acid Extraction

Nucleic acid extraction from 200 microliters of sample (EDTA blood, suspension from oral swab or scabs) was performed using the commercial IndiMag Pathogen Kit (Indical, Leipzig, Germany) in a BioSprint 96 automated extraction system (Qiagen, Hilden, Germany) according to the manufacturer’s instructions.

### 2.5. Capripox Virus Real-Time PCR

In order to detect the Capripox viral genome, real-time PCR (rPCR) was described by Bowden et al., 2008 [11]. The rPCR targets the ORF074 gene, which encodes the P32 protein. Briefly, 2 μL of extracted viral DNA was tested in a 20 μL PCR reaction containing 10 μL of PCR PATH-ID™ qPCR Master Mix (Thermo Fisher Scientific, Waltham, MA, USA), 0.9 μM of each primer (CaPV-074F1/CaPV-074R1), and 0.25 μM of probe CaPV-074P1. The following thermal amplification profile was used: an initial heating step at 95 °C for 15 min, followed by 45 cycles of 15 s at 95 °C and 1 min at 60 °C. Samples were considered positive when a typical amplification curve was obtained and the cycle threshold (Ct) value was lower or equal to 35 (Ct ≤ 35), inconclusive (INC) when 35 < Ct ≤ 40, and negative when no Ct was obtained. For the analysis, INC results are considered as positive.

### 2.6. Sheep Pox Identification by Gel-Based PCR

A gel-based PCR test described by Lamien et al., 2011 [12], was used to differentiate SPPV from GTPV and LSDV by employing two primers targeting the RP030 gene. Briefly, 2 μL of extracted viral DNA was tested in a 25 μL PCR reaction containing 12.5 μL of GoTaq^®^ Hot Start Green Master Mix (Promega, Madison, WI, USA) and 0.6 μM of both primers. The thermal cycling protocol was 95 °C for 2 min followed by 40 cycles of 95 °C for 30 s, 54 °C for 30 s, and 72 °C for 30 s, with a final extension step at 72 °C for 10 min. PCR products were run in an agarose gel using a concentration of 3% at 100 Volts for 1.5 h. Samples positive for SPPV showed an amplicon size of 151 pb (SPPV), while GPPV and LSDV would have an anticipated amplicon size of 172 pb.

### 2.7. Differentiation between Field and Vaccine Strain Sheep Pox Virus

To verify whether the outbreak strain differentiated from the Romania 65 (RM65) vaccine strain, a gel-based PCR that amplifies the ORF25/26 gene was performed [13]. The primers SPPV-Dif1F and SPPV-Dif2R were used in a PCR reaction containing 12.5 μL of GoTaq^®^ Hot Start Green Master Mix (Promega) and 0.8 μM of each primer. The thermal cycling protocol was 95 °C for 2 min followed by 40 cycles of 95 °C for 30 s, 54 °C for 30 s, and 72 °C for 1 min, with a final extension step at 72 °C for 10 min. PCR products were run in an agarose gel using a concentration of 3% at 100 Volts for 1.5 h. The amplicon length for RM65 vaccine strains is expected to be 42 base pairs shorter than that for wild-type SPPV (195 bp for RM65 vaccine strains and 237 bp for wild-type SPPV).

### 2.8. Detection of Specific Antibodies against Capripox Virus by Double Recognition ELISA

The presence of Capripox virus-specific antibodies in sera was assessed using a commercial double recognition ELISA (dr-ELISA) from Innovative Diagnostics. The test was performed as indicated by the manufacturer’s instructions. The S/P percentage of each sample was calculated as follows: S/P% = [(OD Sample − OD Neg Control)/(OD Pos Control − OD Neg Control)] × 100. Samples showing S/P% values ≥ 30% were considered positive; samples with S/P% < 30% were considered negative.

### 2.9. Statistical Analysis

Non-parametric Wilcoxon tests were performed to compare mean Ct values in paired samples in different situations: oral swabs collected at different points post infection or at farms at different points post infection. Comparison between Ct values for unpaired samples was done with the Mann–Whitney test: homogenized scabs vs. storage medium in the index outbreak, individual swabs vs. pooled swabs, scab vs. blood, and oral swab vs. blood in the index case or oral swab in different areas (AND and C-LM). A McNemar test was carried out to assess the correlation between animals found to be SPPV positive (i) in EDTA blood and/or scabs, (ii) in EDTA blood and ELISA, and (iii) scabs and ELISA, originating from the index outbreak. Chi-squared tests were used to compare proportions of positive animals in different tests during repeated samplings in positive farms. For all the analyses, differences were considered significant when the *p*-value was less than 5% (*p* < 0.05). The statistical software used was XLSTAT (V. 2023.3.1.1416).

### 2.10. Ethical Statement

The diagnostic samples collected from sheep and goats analysed in this study were taken from animals as part of veterinary investigations. Further ethical approval was, therefore, not obtained.

## 3. Results

### 3.1. Confirmation of SPPV in the Index Case in Andalusia

On the farm where the first SPPV outbreak was confirmed, clinical signs were observed in more than 50 sheep, including in 30 animals that had already died. The sheep showed generalized skin lesions, including erythema and papules, but several also had scabs present (Figure 2). Therefore, scab samples were taken from 27 out of 60 sampled sheep (Appendix A).

Homogenized scabs and the PBS in which scabs had been stored were analysed separately, and all of them were found to be positive. The mean Ct value in the homogenized scabs (Ct = 18.5) was not significantly lower (Mann–Whitney test; *p* = 0.053) than that of the storage medium (Ct = 20.4) (Figure 3). On the other hand, only 45% (27/60) of collected EDTA blood samples were found to be positive by rPCR, but more interestingly, only 9 out of 27 sheep with SPPV-positive scabs were also positive on EDTA blood samples. No correlation was found between a positive result by rPCR in EDTA blood samples and the presence of scabs (McNemar; *p* = 1.000). Ct values in homogenized scabs and scab storage medium were significantly lower (Mann–Whitney; *p* < 0.0001) than the mean Ct value in EDTA blood samples (Ct = 34.5) (Figure 3). Considering serology, 40% of sampled sheep (24/60) were positive by ELISA (Table 3). No correlation was found between a positive result by ELISA in serum samples and by rPCR in EDTA blood samples (McNemar; *p* = 0.532), nor between the presence of scabs and a positive result by ELISA (McNemar; *p* = 0.590).

Analysis of samples from the first three notified outbreaks in AND and the first five outbreaks notified in C-LM by the gel-based PCR specified by Lamien et al. 2011 [12], showed that the Capripox virus was the Sheep pox virus strain responsible for the outbreaks. In addition, the results obtained using the gel-based PCR described by Haegeman et al. 2015 [13] confirmed that the circulating strain was not a RM65-related vaccine strain. Figure 4 and Figure 5 show the results obtained in samples 121, 122, and 135 corresponding to the index case (outbreak 2022/1).

### 3.2. Overall Clinical and Laboratory Surveillance in Sheep

In AND, after the index case was confirmed, a total of 17 ovine farms were sampled and analysed by rPCR, as suspicious animals were found during the Active surveillance period (Table 1). SPPV outbreaks were confirmed on 12 of those farms, which resulted in a total of 13 outbreaks declared in this area. The clinical signs detected during the Active surveillance period were very mild. Only erythema and sometimes papules were detected in one or a limited number of sheep. During the reinforced active surveillance period in AND, no clinical signs were observed in any of the 49 inspected farms, and none of the 3772 collected oral swabs were found positive by rPCR.

In C-LM, a total of 17 outbreaks were notified. SPPV outbreaks were rPCR-confirmed on 10 out of 11 farms where suspicious sheep were detected during the first Active surveillance period (Table 1), including the farm having an epidemiological link with the first outbreak in AND. In that specific farm, a single sheep with clinical symptoms was observed. The animal was detected due to its reluctance to move, and upon closer inspection, papules were seen on its skin. An oral swab was collected, which was confirmed as positive by rPCR. Similar to AND, the number of sheep with clinical signs was very limited on the farms found to be SPPV-positive during the first Active surveillance period. In line with the limited number of SPPV animals detected on the positive farms, the main reported clinical symptoms in the C-LM area were mild and included fever, depression, and apathy. On a few occasions, swelling of the eyelids and head, rhinitis with nasal discharge, and crusts in the muzzle were also observed. Sometimes, papules and erythema were observed, located mainly in the posterior thirds and spreading rapidly throughout the body; these were best observed in areas lacking wool (Figure 2a,b).

During the reinforced active surveillance period in C-LM, a total of 31.647 oral swab samples were collected, including re-visited farms (Table 1), and 301 (<1%) were found positive by rPCR (Table 4). These positive swabs were collected from four farms where clinically suspected SPPV sheep were detected during the visit by the OVS, but also from two farms without clinical suspicions. However, during subsequent visits of these two farms around 6 days later, laboratory results were confirmed, and animals with mild clinical signs were observed, leading to the notification of outbreaks 2023/2 and 4 (Table 4).

The last outbreak in C-LM (2023/7) was detected in a sheep farm during the second Active surveillance period. Sheep showed mild clinical signs in line with what was observed in other positive farms during the first active surveillance period.

### 3.3. Laboratory Surveillance in Positive Farms

In the AND area, all outbreaks, except the index case, were notified during the Active surveillance period. In this period, EDTA blood samples were collected from multiple sheep in the SPPV-suspected farms, including animals without clinical signs. Percentages of rPCR-positive sheep in EDTA blood samples ranged from 0 to 58% (Table 3). Oral swabs were collected in eight farms but only from animals with clinical symptoms. The percentage of rPCR positivity in these swabs was close to 100%. The Ct values obtained in EDTA blood and oral swab samples were generally high, with 30.6 on average in both kinds of sample (Figure 6). Serum samples were collected on five farms, and seroconversion was only found in a very limited number of animals (less than 2% of sampled sheep) on two farms (Table 3). Additionally, during the second visit of outbreak 2022/18 and 19, serum samples were collected and seropositivity was detected at 16.6% (10/60) and 5% (3/60), respectively.

Sampling during the active surveillance in C-LM strongly relied on the collection of oral swabs (Table 4). Interestingly, the mean Ct value obtained in oral swabs from positive animals with clinical signs in C-LM (Ct = 25.4) was significantly lower (Mann–Whitney test; *p* = 0.003) than that obtained in clinically diseased sheep in AND (Ct = 30.6) (Figure 6). Serum samples were only collected from the revisited 2023/4 farm, where recent skin lesions (papules and erythema) were present, and no seroconversion was detected (Table 4). Additionally, serum samples were collected from three farms that were re-sampled on the day of culling (2022/15 and 22; 2023/7), and percentages of seropositivity of 10% (6/60), 37.5% (3/8), and 16.6% (10/60) were observed.

**Table 4 viruses-16-01034-t004:** Details regarding the type of samples, the laboratory results, and the presence of clinical signs in sheep outbreak farms in Castilla-La Mancha area.

					% (Positive/Total Samples) ^†^			
Outbreak	Period of Surveillance	% Morbidity (Affected/Census)	% Mortality (Death/Census)	Type of Clinical Signs	EDTA Blood	Oral Swab	Scab	Serum
2022/3	Epidemiological link	0.1 (1/890)	0.0 (0/890)	Mild	n.c.	100 (3/3)	n.c.	n.c.
2022/6	Active	0.4 (1/227)	0.0 (0/227)	Mild	n.c.	100 (1/1)	n.c.	n.c.
2022/7	Active	0.4 (8/1.877)	0.0 (0/1.877)	Mild	n.c.	100 (2/2)	n.c.	n.c.
2022/8	Active	0.2 (15/5.075)	0.0 (0/5.075)	Mild	n.c.	100 (2/2)	n.c.	n.c.
2022/4	Active	2.3 (182/7.654)	0.0 (0/7.654)	Mild	n.c.	100 (5/5)	n.c.	n.c.
2022/9	Active	2.5 (15/591)	0.0 (0/591)	Mild	n.c.	100 (2/2)	n.c.	n.c.
2022/15	Active	0.02 (2/7.354)	0.0 (0/7.354)	Mild	n.c.	100 (2/2)	n.c.	n.c.
	Re-visited	no data	no data	Scabs	41.6 (25/60)	80 (48/60)	100 (20/20)	10 (6/60)
2022/16	Active	0.2 (10/3.591)	0.0 (0/3.591)	Mild	n.c.	100 (2/2)	n.c.	n.c.
2022/22	Active	0.9 (15/1.519)	0.3 (5/1.519)	Mild	n.c.	100 (9/9)	n.c.	n.c.
	Re-visited	no data	no data	Mild	62.5 (5/8)	n.c.	n.c.	37.5 (3/8)
2022/23	Active	3.6 (30/820)	1.8 (15/820)	Mild	n.c.	100 (9/9)	n.c.	n.c.
2023/1 *	Reinforced	3.6 (50/1.359)	0.0 (0/1.359)	Mild	n.c.	88.8 (8/9)	n.c.	n.c.
2023/2	Reinforced	0.0 (0/3.544)	0.0 (0/3.544)	No	n.c.	5–24 (7/145) **	n.c.	n.c.
	Revisited	0.02 (1/3.544)	0.0 (0/3.544)	Mild	n.c.	63.8 (23/36)	n.c.	n.c.
2023/3	Reinforced	5.9 (480/8.100)	0.06 (5/8.100)	Mild	n.c.	20–100 (30/147) **	n.c.	n.c.
2023/4	Reinforced	0.0 (0/1.216)	0.0 (0/1.216)	No	n.c.	16–79 (9/57) **	n.c.	n.c.
	Revisited	0.3 (4/1.216)	0.0 (0/1.216)	Mild	n.c.	20 (12/60)	n.c.	0 (0/60)
2023/5	Reinforced	0.3 (5/1.410)	0.0 (0/1.410)	Mild	n.c.	26.6 (16/60)	n.c.	n.c.
2023/6 *	Reinforced	0.03 (1/3.142)	0.0 (0/3.142)	Mild	n.c.	100 (2/2)	n.c.	n.c.
	Revisited	no data	no data	Mild	n.c.	1.3–6.8 (2/145) **	n.c.	n.c.
2023/7 *	Active	6.2 (21/334)	0.5 (2/334)	Mild	n.c.	100 (21/21)	n.c.	n.c.
	Revisited	no data	no data	Scabs	45 (27/60)	98.3 (59/60)	100 (16/16)	16.6 (10/60)

n.c.: not collected; ^†^ Serum samples were analysed by ELISA. Oral swab, EDTA blood, and scabs were analysed by qPCR; * The farm also had goats without clinical signs that were sampled, and all were negative by rPCR and/or ELISA; ** Analysed in pools (5 samples).

### 3.4. Evolution of SPPV Infection in Affected Farms

Seven outbreak farms, three from AND (2022/12, 2022/18, and 2022/19) and four from C-LM (2022/15, 2022/22, 2023/6, and 2023/7) were re-sampled the day that sheep were culled, a few days after the first sampling. Clinical signs observed during the first sampling were mild and mainly limited to erythema, while some sheep from four outbreak farms (2022/15, 18, and 19; 2023/7) had developed scabs by the time of the second sampling. In addition to the aggregated results at the farm level described in Table 3 and Table 4, a substantial part of both samplings was done on the same animals (Appendix A), enabling some individual comparisons.

In outbreak farm 2022/18, the percentage of positive oral swab samples had significantly decreased (Chi-squared test; *p* < 0.001) from 97.6% (42/43) to 51.6% (31/60). No significant decrease (from 100% (21/21) to 98.3% (59/60); Chi-squared test; *p* = 0.552) was seen in outbreak farm 2023/7, which was resampled after 6 days. As a result of this 6-day interval, the Ct value in oral swabs from 34 sheep that were positive in both samples significantly increased (Wilcoxon test; *p* = 0.002) between the first and second sample collection, from 27.3 to 30.4 (Figure 7).

Regarding EDTA blood samples in outbreak farms where scabs were observed, the percentage of positive EDTA blood slightly increased from 32% (16/50) to 40% (24/60) in outbreak farm 2022/18 and decreased from 47.7% (32/67) to 36.6% (22/60) in outbreak farm 2022/19, respectively, without being statistically significant (Chi-squared test; *p* = 0.385 and *p* = 0.207). Ct values in the two farms increased from 32.6 to 34.3 and from 31 to 32.7, respectively (Figure 8), although this was only statistically significant in the outbreak farm 2022/18 (Mann–Whitney; *p* < 0.0001 and *p* = 0.107, respectively). Interestingly, in outbreak farm 2022/19, just 1 out of 29 sheep was EDTA blood-positive in both samplings. On the other hand, in outbreak farm 2022/12, where no scabs were observed, the Ct value in EDTA blood samples significantly decreased on average from 30.9 to 28.5 (Mann–Whitney; *p* = 0.047) (Figure 8). Additionally, the percentage of positive EDTA blood significantly increased from 43% (26/60) to 76.1% (16/21) (Chi-squared test; *p* = 0.009) over a 6-day interval.

Serological data could only be compared in outbreak farm 2022/12, where a significant increase from 1.6% (1/60) to 19% (4/21) between samplings was observed (Chi-squared; *p* = 0.004).

### 3.5. SPPV Detection in Goat Samples

Goats were clinically inspected and sampled when mixed sheep and goat farms were found positive for SPPV or when they were located close to positive sheep farms (Table 2). No clinical signs were observed in any goat on the 30 farms included, and all 1757 oral swab, 354 EDTA blood, and 167 serum samples were negative in rPCR and ELISA.

### 3.6. Comparative SPPV Detection in Blood and Oral Swabs Early after Infection

Appendix A shows rPCR results from 229 paired EDTA blood and oral swab samples collected from 212 sheep without scabs from outbreak farm 2022/12, 15, 18, and 19. They were analysed to determine if one of both matrices was more suitable for SPPV detection early after infection. If oral swabs were considered as the golden standard, the relative sensitivity of EDTA blood compared to swabs was only 48,9% (42.1–55.9), showing that testing blood results in many false negative results (Table 5). These results confirm the use of oral swabs as the sample of choice for Capripox virus genome detection early after infection. In addition, the mean Ct value in oral swabs (Ct 27.1 ± 4.3) of the 91 sheep positive in both matrices was significantly lower (Mann–Whitney test; *p* < 0.0001) than the mean Ct in EDTA blood (Ct 31.5 ± 3.7). A scatter plot showing corresponding Ct values in oral swabs and EDTA blood can be found in Figure 9.

### 3.7. Effect of Oral Swab Pooling on Sensitivity

The promising results of oral swabs as a diagnostic matrix described above were obtained while the outbreak was ongoing. Based on the extensive number of swabs that was planned to be collected during the reinforced active surveillance period and the associated analysis costs, the effect of pooling five swabs on the sensitivity of SPPV detection was evaluated. Two positive farms were therefore resampled, and oral swabs from 60 sheep from each farm were collected in duplicate (replicate A and B). The 120 replicate A samples were individually eluted in 1 mL of PBS and analysed. Seventy-nine of them tested positive, and 41 samples were negative. Afterwards, the replicate B swabs corresponding to 32 positive animals were each analysed in a pool of five, prepared by eluting one replicate B swab together with four negative oral swabs in 2 mL of PBS. The negative swabs were collected from healthy animals from an SPPV-free area and were confirmed as negative by rPCR. From the 32 oral swabs tested in the pool, 28 were also found positive in rPCR. Four swabs were found negative when tested in a pool of five. Their corresponding paired swabs had Ct values of 35.1, 29.9, 36.2, and 31.5. This corresponds to a relative sensitivity of 87.5% for swab testing in pools compared to individual swab testing. The mean Ct value from the 28 positive swabs tested in pools (Ct = 29.4) was not significantly different (Mann–Whitney test; *p* = 0.528) from the mean Ct value (Ct = 29.9) obtained in their corresponding paired swabs, which were analysed individually (replicate A) (Figure 10).

### 3.8. Laboratory Testing during Repopulation

No clinical signs compatible with an SPPV infection were observed in any animal after repopulation on the 14 repopulated farms. On 13 of the repopulated farms, all samples collected after a few days of repopulation, being serum and oral swab samples, were negative by ELISA and rPCR, respectively. On one farm, however, two out of sixty oral swabs were found positive in rPCR, while all animals were seronegative. That farm was revisited and re-sampled four more times. Clinical signs were not observed in any animal, and all 230 collected serum samples were negative in ELISA. However, a few oral swabs were found positive by rPCR during all visits (Ct = 35.9 ± 0.8). During the last visit, 10 environmental samples were also collected at the farm, and one was found positive in rPCR (Ct = 33.7) (Table 6). 

## 4. Discussion

Despite the close proximity of sheep pox virus in Northern Africa [8], Spain has remained free of the disease for more than 50 years. However, in 2022 sheep pox was identified and confirmed by the NRL in Spain in two areas, namely Andalusia and Castilla-La Mancha. At the moment of first detection, the disease had already progressed significantly in the affected herd, as was witnessed by a large number of dead sheep and sheep with scabs. The latter sample type proved to be the most ideal sample to be collected, as the Ct values are very low and provide an 100% detection rate. The high viral load in scabs does however represent a challenge in preventing sample and laboratory contamination. Another advantage of scabs is that they remain present for more than 6 weeks on the animals [14], and Capripox viruses have been demonstrated to be very stable in this kind of biological matrix [15]. This allows for a large window in which the detection of the virus can be achieved. However, scabs take time to form, as they first pass through the erythema, macula, and papule stages. This becomes even more important in situations where the disease has a mild course. This is reflected by the findings during the active surveillance in Spain, whereby only erythema or papules were noted at the moment of SPPV confirmation in the infected herds. Therefore, it can be stated that scabs are the most interesting sample to take if present but are less suited for early detection.

Scabs can be collected dry or in a transport medium. Although the Ct value of the scabs itself is lower than the transport medium, the latter remains interesting, as some DNA extraction protocols of tissues can be more elaborate or take longer than those for liquids. Indirectly, the transfer of the virus from the scab to the transport medium shows also that liquids in contact with a scab-like material can become strongly positive and therefore represent a potential transmission pathway.

In contrast to scabs, detection using EDTA blood requires a viremia, which is more limited in time. In experimental challenge studies, the viremia disappears after approximately 2/3 weeks after challenge [11,16]. This is reflected by the field samples in this study, where only 9 out of 27 animals with scabs in outbreak farm 2022/1 were positive in EDTA blood, or in resampled outbreak farm 2022/19, where only 1 out of 29 sheep was EDTA blood-positive in both samplings.

The analysis of the paired blood/oral swab samples from animals without scabs demonstrated the usefulness of oral swabs. The higher positivity of the oral swabs compared to blood could allow an increased detection sensitivity of an outbreak in the herd. These field data are substantiated by experimental studies where SPPV could be longer detected in oral swabs in contrast to blood [11,16]. Similarly, a higher sensitivity in nasal swabs, compared to buffy coat, was reported by Balinsky et al., 2008 [5]. Additionally, resampled outbreak farms 2022/18 and 2023/7 showed lower percentages of oral swab positivity and higher Ct values after scab lesions appeared. Therefore, the oral swab seems to be a relevant sample for early detection and complementary to scabs.

One should, however, remain careful when using swabs to determine the infection status of individual animals, as positive swabs could also be the result of environmental contamination. This was seen during the outbreak in Spain after repopulation of one of the SPPV-infected farms after it was cleaned and disinfected. This, however, does not pose an issue when swabs are used for early detection of a SPPV outbreak in a farm and to determine the infection status on a herd level and highlights the importance of observing clinical signs compatible with the disease. Furthermore, our data showing the capacity to pool swabs per five with only a slight loss of sensitivity further increases the usefulness of this sample type. Another benefit is that swabs are more easily collectable than blood samples and that their collection is non-invasive, thereby reducing the risk of secondary infections.

Data obtained during this outbreak confirmed that serology is not a useful tool for early detection. The latter is caused by the fact that an antibody response needs time to develop and that not all sheep develop an antibody level that is above the limit of detection of the serological tests available. This is shown by the results of positive farms before observing scabs, where no seropositivity or a very low percentage (<2%) was detected. Even in outbreak farm 2022/1, where a consolidated infection with severely clinically diseased sheep took place, only 40% of sampled animals were positive in ELISA. Even sheep with clear skin lesions were negative in ELISA. This is in line with previous studies looking at seroconversion after capripox challenge, where the onset of seroconversion could take more than 2 weeks and even be transient [16,17,18].

Upon detection of the first SPPV-positive farms in AND and C-LM, an extensive active surveillance was implemented based on clinical inspection complemented with sampling for laboratory analysis. The clinical inspection during the first weeks allowed us to detect several outbreaks, which were confirmed in the laboratory. The type of lesion observed and the limited number of affected animals on farms that were confirmed SPPV-positive suggest that active surveillance systems based on clinical inspection allowed an early detection of outbreaks. Nevertheless, since the outbreak was not rapidly contained, it was decided to reinforce the active surveillance by collecting oral swabs at every inspected farm, even if no clinical signs were detected, to improve the early detection of cases. Both the sampling by the veterinarians and the analysis of samples in the laboratory required an enormous effort. More than 6000 and 35,000 samples were analysed in AND and C-LM areas, respectively. This was the first time that a surveillance based on massive laboratory analysis by qPCR of swab samples is reported. This intensive laboratory surveillance effort allowed the early detection of the SPPV infection in two farms, before any clinical signs were observed. In both cases, clinical signs had developed in some animals when the farm was revisited around 5 days later. It must thus be assessed whether the benefit of advancing an SPPV detection on a limited number of farms thanks to the intensive laboratory surveillance justifies the economic cost of such an intense sampling and laboratory analysis. The Spanish government decided to return to the normal active surveillance from 31 March 2023 onwards, consisting of clinical inspection complemented only with sampling for laboratory analysis in the case that suspected clinical signs were observed.

In the literature, certain sheep pox virus strains have been reported to also cause disease in goats [19,20]. This, however, was not the case for the SPPV strain responsible for the outbreak in Spain. No clinical signs nor positive serological (ELISA) or virological (PCR) results were found in goats during this SPPV outbreak, despite the intense sampling on mixed farms where SPPV-positive sheep lived with goats. This Spanish strain thus seems to have a strong host specificity for sheep, something that was previously also already reported for several other strains, e.g., the strain responsible for the SPPV outbreak in Mongolia in 2006–2007 [21].

Multiple farmers who lost their entire flock due to the SPPV outbreak decided to restart their activities a few months later. Fourteen farms were repopulated at least 4,5 months after the official cleaning and disinfection. A thorough clinical follow-up after repopulation proved to be crucial, since some oral swabs collected from the newly introduced sheep were found to be SPPV-positive in the absence of clinical symptoms. The lack of clinical symptoms and absence of a productive SPPV infection were confirmed by the lack of seroconversion. The finding of some SPPV-positive environmental swabs collected after repopulation suggests that the positive oral swabs were caused by contact with a non-infectious virus or viral genome that remained after cleaning and disinfection. This finding should be taken into account when designing repopulation and surveillance protocols. Using sentinel animals that are closely followed by clinical inspection and serology seems to be the best strategy to ensure a safe repopulation.

## Figures and Tables

**Figure 1 viruses-16-01034-f001:**
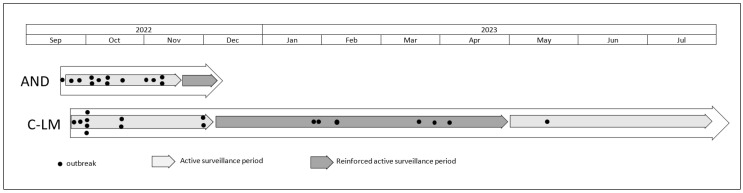
Schedule of sheep pox outbreaks and type of active surveillance applied in the restricted area. AND: Andalusia; C-LM: Castilla-La Mancha.

**Figure 2 viruses-16-01034-f002:**
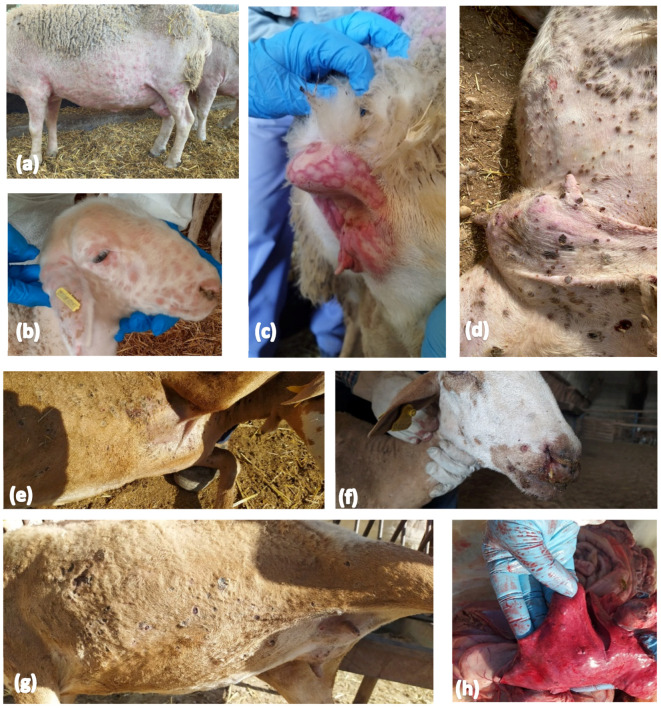
Photographs of skin lesions from Spanish outbreak. (**a**) Erythema; (**b**) papules; (**c**) erythematous papules; (**d**) scabs; (**e**) scabs; (**f**) crusts in the muzzle; (**g**) scabs; (**h**) lesions in lungs. Images (**e**–**h**) correspond to the index case.

**Figure 3 viruses-16-01034-f003:**
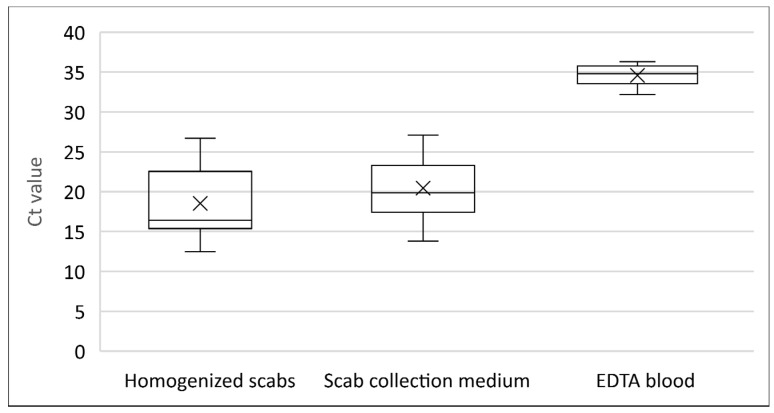
Ct values in different diagnostic samples in the index outbreak (2022/1).

**Figure 4 viruses-16-01034-f004:**
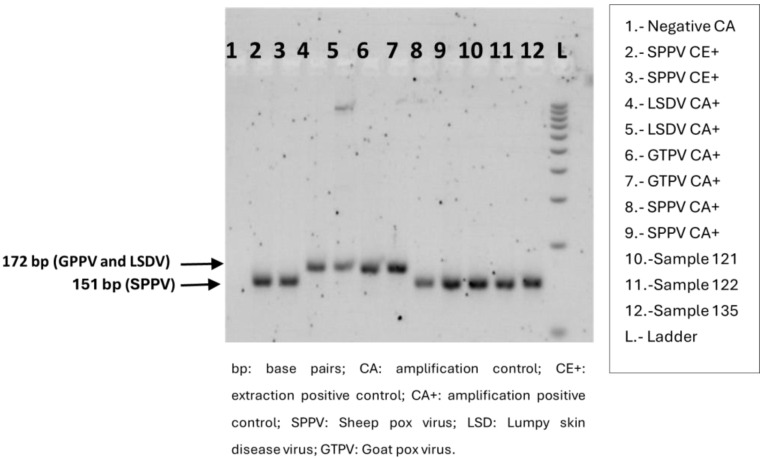
Gel-based PCR for differentiation of capripox virus by amplicon size (Lamien et al., 2011). Samples 121, 122, and 135 are three homogenized scab samples from the index case (outbreak 2022/1).

**Figure 5 viruses-16-01034-f005:**
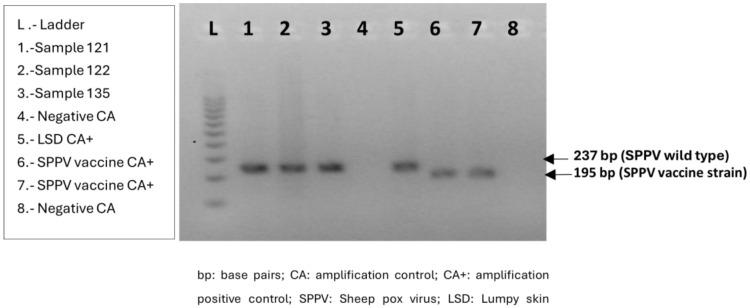
Gel-based PCR for differentiation of vaccine (RM65) versus field strains (Haegeman et al. 2015). Samples 121, 122, and 135 are three homogenized scab samples from the index case (outbreak 2022/1).

**Figure 6 viruses-16-01034-f006:**
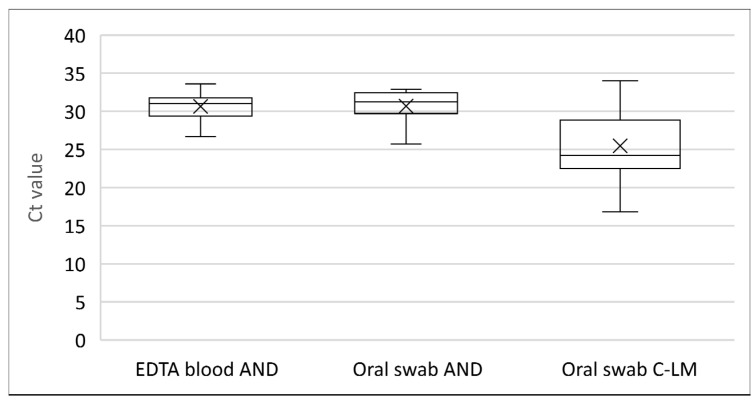
Ct value comparison from positive farms in Andalusia (AND) and Castilla-La Mancha (C-LM).

**Figure 7 viruses-16-01034-f007:**
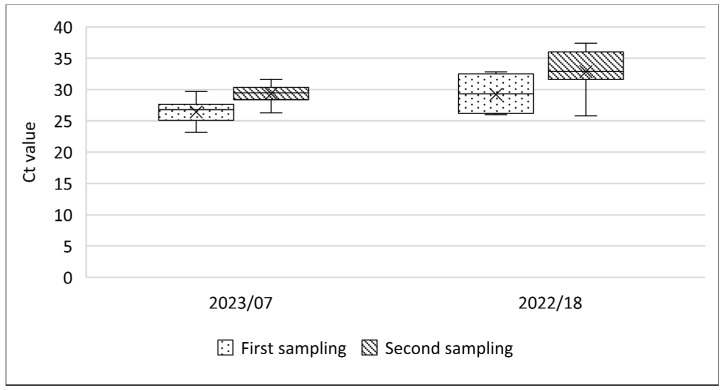
Evolution of Ct values in oral swabs from outbreak farms where scabs were observed in the second sampling.

**Figure 8 viruses-16-01034-f008:**
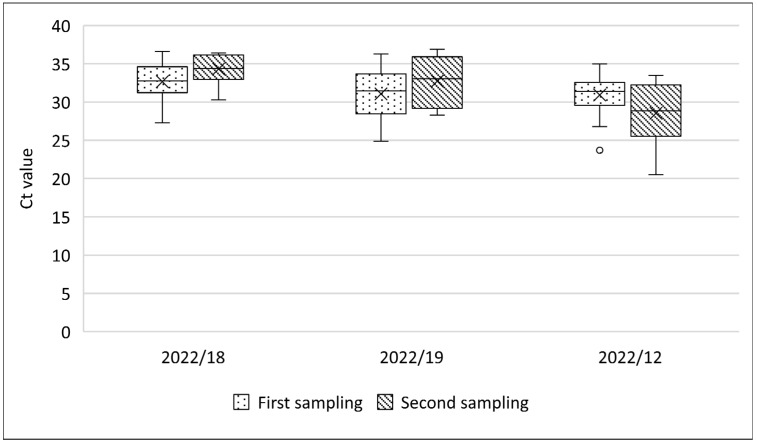
Evolution of Ct values in EDTA blood from outbreak farms where scabs were observed during the second sampling (2022/18 and 19) or not (2022/12).

**Figure 9 viruses-16-01034-f009:**
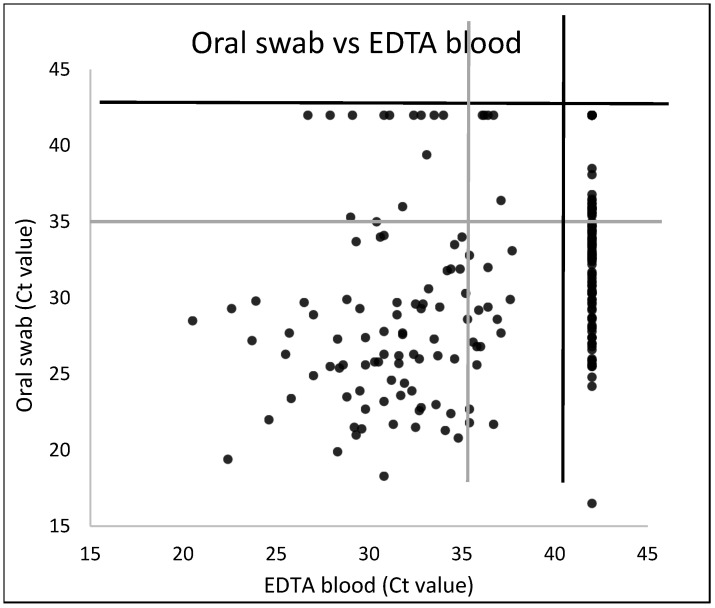
Scatter plot showing Ct values in 229 paired EDTA blood and oral swab samples.

**Figure 10 viruses-16-01034-f010:**
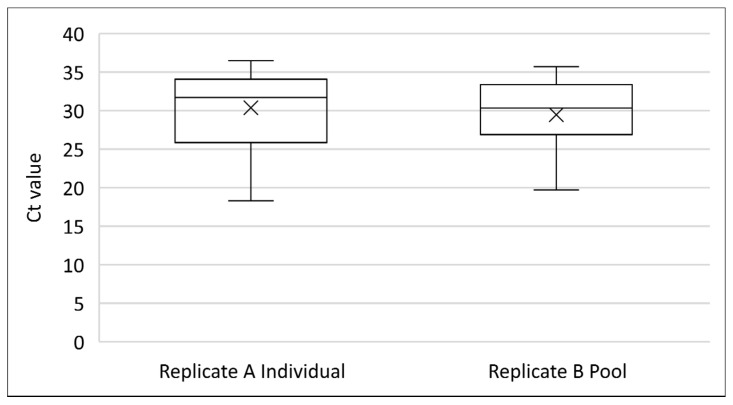
Comparison between Ct values obtained for individual (replicate A) and pooled (replicate B) oral swabs.

**Table 1 viruses-16-01034-t001:** Sheep farms officially sampled in the restricted areas.

Period	Area	Number of Farms	Number of Samples			
		(Positive/Sampled)	Oral Swab	EDTA Blood	Scab	Serum
First period of Active surveillance	AND	12/17	309	817	11	372
	C-LM	10/11 ^†^	189	68	20	0
Reinforced active surveillance	AND	0/49	3.772	0	0	0
	C-LM	6/286	31.647	10	17	60
Second period of Active surveillance	AND	0	n.c.	n.c.	n.c.	n.c.
	C-LM	1/25	470	60	16	60
		29/388	36,387	955	64	492

AND: Andalusia; C-LM: Castilla-La Mancha; n.c.: not collected; ^†^ Included first outbreak detected by epidemiological link to outbreak in AND.

**Table 2 viruses-16-01034-t002:** Goat farms officially sampled in the restricted areas.

Period	Area	Number of Farms	Number of Samples			
		Sampled	Oral Swab	EDTA Blood	Scab	Serum
First period of Active surveillance	AND	10	102	297	0	124
	C-LM	0	n.c.	n.c.	n.c.	n.c.
Reinforced active surveillance	AND	7	173	57	0	0
	C-LM	12	1479	0	0	43
Second period of Active surveillance	AND	0	n.c.	n.c.	n.c.	n.c.
	C-LM	1	3	0	0	0
		30	1757	354	0	167

AND: Andalusia; C-LM: Castilla-La Mancha; n.c.: not collected.

**Table 3 viruses-16-01034-t003:** Details regarding the type of samples, the laboratory results, and the presence of clinical signs in sheep outbreak farms in the Andalusia area.

					% (Positive/Total Samples) ^†^			
Outbreak	Period of Surveillance	% Morbidity (Affected/Census)	% Mortality (Death/Census)	Type of Clinical Signs	EDTA Blood	Oral Swab	Scab	Serum
2022/1 *	Index case (passive)	15.9 (50/314)	9.5 (30/314)	scabs	45 (27/60)	n.c.	100 (27/27)	40 (24/60)
2022/2 *	Active	0.5 (1/170)	0.0 (0/170)	mild	1.7 (1/58)	n.c.	n.c.	1.7 (1/58)
2022/5 *	Active	0.5 (2/340)	0.0 (0/340)	mild	3.3 (2/59)	100 (2/2)	n.c.	0 (0/59)
2022/10 *	Active	2.7 (3/110)	0.0 (0/110)	mild	3.9 (2/51)	100 (3/3)	n.c.	0 (0/51)
2022/11 *	Active	1.2 (1/79)	0.0 (0/79)	mild	0 (0/38)	100 (1/1)	n.c.	0 (0/38)
2022/12 *	Active	1.0 (7/639)	0.0 (0/639)	mild	43 (26/60)	100 (4/4)	n.c.	1.6 (1/60)
	Re-visited	no data	no data	mild	76.1 (16/21)	100 (21/21)	n.c.	19 (4/21)
2022/13 *	Active	26 (50/192)	0.0 (0/192)	mild	58.4 (31/53)	n.c.	n.c.	n.c.
2022/14	Active	9.5 (4/42)	0.0 (0/42)	mild	44 (11/25)	n.c.	n.c.	n.c.
2022/17	Active	0.2 (1/373)	0.0 (0/373)	mild	15 (9/60)	76.9 (10/13)	n.c.	n.c.
2022/18 *	Active	15.4 (15/97)	0.0 (0/97)	mild	32 (16/50)	97.6 (42/43)	n.c.	n.c.
	Re-visited	no data	no data	scabs	40 (24/60)	51.6 (31/60)	100 (9/9)	16.6 (10/60)
2022/19 *	Active	8.2 (30/364)	0.0 (0/364)	mild	47.7 (32/67)	100 (7/7)	n.c.	n.c.
	Re-visited	no data	no data	scabs	36.6 (22/60)	96.6 (58/60)	100 (2/2)	5 (3/60)
2022/20 *	Active	9.7 (20/206)	0.0 (0/206)	mild	52 (26/50)	n.c.	n.c.	n.c.
2022/21 *	Active	0.6 (1/149)	0.0 (0/149)	mild	17.3 (9/52)	40 (2/5)	n.c.	n.c.

n.c.: not collected; ^†^ Serum samples were analysed by ELISA. Oral swabs, EDTA blood, and scabs were analysed by qPCR; * The farm also had goats without clinical signs that were sampled, and all were negative by PCR and/or ELISA.

**Table 5 viruses-16-01034-t005:** Contingency table showing rPCR results obtained in paired EDTA blood and oral swab samples collected from 212 sheep (17 were sampled twice).

		EDTA Blood	
		Positive or Inconclusive	Negative
Oral swab	Positive or inconclusive	91	97
	Negative	13	28

**Table 6 viruses-16-01034-t006:** Laboratory results in repopulated farms.

		% (Positive or Inconclusive/Total Samples)		
Outbreak	Interval to Repopulation (Days) *	Oral Swab	Serum	Environmental Swab
2022/3	154	0 (0/60)	0 (0/60)	n.c.
2022/4	153	0 (0/60)	0 (0/60)	n.c.
2022/6	150	0 (0/60)	0 (0/60)	n.c.
2022/7	150	3.3% (2/60)	0 (0/60)	n.c.
	156	3.5% (2/57)	n.c.	n.c.
	167	8.7% (5/57)	0 (0/57)	n.c.
	182	12.2% (7/57)	0 (0/57)	n.c.
	202	n.c.	0 (0/56)	10% (1/10)
2022/8	150	0 (0/60)	0 (0/60)	n.c.
2022/9	153	0 (0/60)	n.c.	n.c.
2022/10	428	0 (0/31)	0 (0/31)	n.c.
2022/15	140	0 (0/60)	0 (0/60)	n.c.
2022/17	274	0 (0/08)	n.c.	n.c.
2022/20	369	0 (0/37)	n.c.	n.c.
2022/22	305	0 (0/47)	n.c.	n.c.
2023/2	185	0 (0/43)	n.c.	n.c.
2023/3	265	0 (0/91)	n.c.	n.c.
2023/6	189	0 (0/94)	n.c.	n.c.
		(16/942)	(0/561)	(1/10)

* Time (days) between cleaning and disinfection and repopulation; n.c.: not collected.

## Data Availability

The data that support the findings of this study are available from the corresponding author upon reasonable request.

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
