# Peer review of "Lessons Learned from Active Clinical and Laboratory Surveillance during the Sheep Pox Virus Outbreak in Spain, 2022–2023"

_viruses, 2024, doi:10.3390/v16071034_

Round 1

Reviewer 1 Report

Comments and Suggestions for Authors

The manuscript analyzes sheeppox outbreaks in Spain in 2022-2023. The authors conducted a large amount of research. As a result, an alternative approach for early diagnosis of the disease is proposed. Detection of the viral genome in oral lavages using PCR showed a higher sensitivity compared to detection of the viral genome in blood. Viral isolation will be required in some cases to confirm the diagnosis.

Minor:

Figure 1 - expand abbreviations as in table 1 (AND, C-LM);

Figure 4 - Revise the legend. There is no clear description of the samples analyzed.

Figure 5 - It is recommended to present a clearer (in focus) image if possible

Author Response

Comment 1: The manuscript analyzes sheeppox outbreaks in Spain in 2022-2023. The authors conducted a large amount of research. As a result, an alternative approach for early diagnosis of the disease is proposed. Detection of the viral genome in oral lavages using PCR showed a higher sensitivity compared to detection of the viral genome in blood. Viral isolation will be required in some cases to confirm the diagnosis.

Response 1: Thank you for your comments. Regarding virus isolation, although it was carried out on some samples from first and last outbreaks, it has not been reported in this manuscript since virus isolation has not been used to surveillance.

Comment 2: Figure 1 - expand abbreviations as in table 1 (AND, C-LM); Figure 4 - Revise the legend. There is no clear description of the samples analyzed; Figure 5 - It is recommended to present a clearer (in focus) image if possible

Response 2: comment has been considered. 

Reviewer 2 Report

Comments and Suggestions for Authors

The manuscript is interesting and reports details on an important event related to the spread of the SPV in Spain and the efforts to survey and control it. However, the manuscript contains some inaccuracies that affect overall clarity and need to be addressed before publication.

  1. In lines 4-5, the coauthors' affiliations include an apex number 4 that does not correspond to any of the institutions involved in the research. Please amend this.
  2. In the Introduction section, lines 69-71, the manuscript mentions that a detailed description of the epidemiological findings and control measures is currently being prepared for another publication. Could the authors provide more information about this other article to ensure there is no duplicate publication?
  3. In the Introduction section, Figure 1 shows the end of the reinforced active surveillance period in C-LM according to the timetable reported in the figure. This does not correspond to the date reported in the manuscript, which is the 31st of March 2023. Please ensure these dates are consistent.
  4. Tables 3 and 4 are not very clear. It is difficult to understand how many outbreaks were reported in the two regions. The tables suggest 13 in AND and 17 in C-LM, but this is not reflected in the text. Additionally, the numbers of oral swabs analyzed in the two regions seem to differ between the tables and the manuscript (lines 263 and 279). Please review the result data carefully and make it clearer.
  5. In Figure 4, there is a lack of correspondence between the samples loaded in the gel and the list provided. Please make this alignment clearer.

Author Response

Thank you for your comments. All of them have been considered and it has improved  comprehension of the text and corrected some mistakes.

Comment 1: In lines 4-5, the coauthors' affiliations include an apex number 4 that does not correspond to any of the institutions involved in the research. Please amend this.

Respond 1: mistake corrected

Comment 2: In the Introduction section, lines 69-71, the manuscript mentions that a detailed description of the epidemiological findings and control measures is currently being prepared for another publication. Could the authors provide more information about this other article to ensure there is no duplicate publication?

Respond 2: that manuscript is currently under review. It describes and evaluates the control measures taken by veterinary authorities, but does not describe laboratory results or surveillance activities. The sentence has been changed to explain the objective of that manuscript.

Comment 3: In the Introduction section, Figure 1 shows the end of the reinforced active surveillance period in C-LM according to the timetable reported in the figure. This does not correspond to the date reported in the manuscript, which is the 31st of March 2023. Please ensure these dates are consistent.

Respond 3: mistake corrected. Reinforced active surveillance in C-LM finished on April 27th 2023.

Comment 4: Tables 3 and 4 are not very clear. It is difficult to understand how many outbreaks were reported in the two regions. The tables suggest 13 in AND and 17 in C-LM, but this is not reflected in the text. Additionally, the numbers of oral swabs analyzed in the two regions seem to differ between the tables and the manuscript (lines 263 and 279). Please review the result data carefully and make it clearer.

Respond 4: point 3.2 has been modified in order to clarify the data in tables 13 and 14. Additionally, They have been slightly modified so that the information for each outbreak is in the same row.

Comment 5: In Figure 4, there is a lack of correspondence between the samples loaded in the gel and the list provided. Please make this alignment clearer.

Respond 5: The figures underwent changes when it moved to the magazine format. Both figures have been modified so that the description and result of each gel sample is clear.

Round 2

Reviewer 2 Report

Comments and Suggestions for Authors

The reviewed manuscript addresses the issues raised during peer review